# Clinical Impact of Atypical Chest Pain and Diabetes Mellitus in Patients with Acute Myocardial Infarction from Prospective KAMIR-NIH Registry

**DOI:** 10.3390/jcm9020505

**Published:** 2020-02-12

**Authors:** Jun-Won Lee, Jin Sil Moon, Dae Ryong Kang, Sang Jun Lee, Jung-Woo Son, Young Jin Youn, Sung Gyun Ahn, Min-Soo Ahn, Jang-Young Kim, Byung-Su Yoo, Seung-Hwan Lee, Ju Han Kim, Myung Ho Jeong, Jong-Seon Park, Shung Chull Chae, Seung Ho Hur, Myeng-Chan Cho, Seung Woon Rha, Kwang Soo Cha, Jei Keon Chae, Dong-Ju Choi, In Whan Seong, Seok Kyu Oh, Jin Yong Hwang, Junghan Yoon

**Affiliations:** 1Department of Internal Medicine, Division of Cardiology, Yonsei University Wonju College of Medicine, Wonju 26426, Korea; ljwcardio@yonsei.ac.kr (J.-W.L.); sang3621@yonsei.ac.kr (S.J.L.); sonjwoo@yonsei.ac.kr (J.-W.S.); younyj@yonsei.ac.kr (Y.J.Y.); sgahn@yonsei.ac.kr (S.G.A.); heartsaver@yonsei.ac.kr (M.-S.A.); kimjy@yonsei.ac.kr (J.-Y.K.); yubs@yonsei.ac.kr (B.-S.Y.); carshlee@yonsei.ac.kr (S.-H.L.); 2Center of Biomedical Data Science, Yonsei University Wonju College of Medicine, Wonju 26426, Korea; 3Department of Internal Medicine and Heart Center, Chonnam National University Hospital, Gwangju 61469, Korea; kim@zuhan.com (J.H.K.); myungho@chollian.net (M.H.J.); 4Department of Internal Medicine, Division of Cardiology, Yeungnam University Medical Center, Yeungnam University College of Medicine, Daegu 42415, Korea; pjs@med.yu.ac.kr; 5Department of Internal Medicine, Kyungpook National University Hospital, Kyungpook National University, School of Medicine, Daegu 41944, Korea; scchae@knu.ac.kr; 6Department of Internal Medicine, Keimyung University Dongsan Medical Center, Daegu 42601, Korea; shur@dsmc.or.kr; 7Department of Internal Medicine, College of Medicine, Chungbuk National University, Cheongju 28644, Korea; mccho@cbnu.ac.kr; 8Cardiovascular Center, Korea University Guro Hospital, Seoul 08308, Korea; swrha617@yahoo.co.kr; 9Department of Internal Medicine, Pusan National University Hospital, Busan 49241, Korea; chakws1@hanmail.net; 10Department of Internal Medicine, Chunbuk National University School of Medicine, Division of Cardiology, Jeonju 54907, Korea; jkchae@chonbuk.ac.kr; 11Department of Internal Medicine, Seoul National University Bundang Hospital, Seongnam 13620, Korea; djchoi@snu.ac.kr; 12Department of Internal Medicine, College of Medicine, Chungnam National University Hospital, Chungnam National University, Daejeon 35015, Korea; iwseong@cnu.ac.kr; 13Department of Internal Medicine, Wonkwang University School of Medicine, Division of Cardiology, Iksan 54538, Korea; oskcar@wku.ac.kr; 14Department of Internal Medicine, Gyungsang National University School of Medicine, Gyungsang National University Hospital, Jinju 52727, Korea; jyhwang@gnu.ac.kr

**Keywords:** chest pain, diabetes, myocardial infarction

## Abstract

Atypical chest pain and diabetic autonomic neuropathy attract less clinical attention, leading to underdiagnosis and delayed treatment. To evaluate the long-term clinical impact of atypical chest pain and diabetes mellitus (DM), we categorized 11,159 patients with acute myocardial infarction (AMI) from the Korea AMI-National Institutes of Health between November 2011 and December 2015 into four groups (atypical DM, atypical non-DM, typical DM, and typical non-DM). The primary endpoint was defined as patient-oriented composite endpoint (POCE) at 2 years including all-cause death, any myocardial infarction (MI), and any revascularization. Patients with atypical chest pain showed higher 2-year mortality than those with typical chest pain in both DM (29.5% vs. 11.4%, *p* < 0.0001) and non-DM (20.4% vs. 6.3%, *p* < 0.0001) groups. The atypical DM group had the highest risks of POCE (hazard ratio (HR) 1.76, 95% confidence interval (CI) 1.48–2.10), all-cause death (HR 2.23, 95% CI 1.80–2.76) and any MI (HR 2.34, 95% CI 1.51–3.64) in the adjusted model. In conclusion, atypical chest pain was significantly associated with mortality in patients with AMI. Among four groups, the atypical DM group showed the worst clinical outcomes at 2 years. Application of rapid rule in/out AMI protocols would be beneficial to improve clinical outcomes.

## 1. Introduction

Chest pain is one of the cardinal symptoms that warrants a visit to the emergency department (ED) [1]. Among patients with chest pain, those with pain of cardiac origin consist of a quarter to a third of the total burden [1,2]. It is challenging to determine whether chest pain is related to acute coronary syndrome (ACS). Careful history taking is the first step to distinguish high-risk patients, reduce overcrowding of ED, and prevent unnecessary examinations to control medical expenditure [3,4]. The atypical features of chest pain of non-cardiac origin are considered less urgent and tend to be overlooked and underestimated. Consequently, delayed diagnosis and treatment of these patients, who were eventually diagnosed with acute myocardial infarction (AMI), have known to be associated with unfavorable clinical outcomes [2,5,6,7].

Diabetes mellitus (DM) is considered to have a risk equivalent to that of coronary heart disease, and global prevalence is gradually increasing [8,9]. Diabetic cardiovascular autonomic neuropathy contributes to inappropriate perception of chest pain [10]. A previous study including patients with suspected stable angina showed that the incidence rate of coronary events was higher among patients with typical chest pain, especially among diabetic patients, than that among patients with atypical chest pain [11]. Interestingly, atypical patients with diabetes experienced more coronary events than those without diabetes. Although the recent improvement in optimal medical treatment and device therapy has reduced cardiovascular events, better clinical outcomes of atypical chest pain in patients with AMI are still not established. Therefore, this study aimed to assess the long-term impact of atypical chest pain in AMI patients with or without DM.

## 2. Methods

### 2.1. Study Population

The Korea Acute Myocardial Infarction-National Institutes of Health registry (KAMIR-NIH) is a prospective, multicenter, nationwide observational cohort study (cris.nih.go.kr identifier: KCT-0000863). Patients diagnosed with AMI in 20 tertiary hospitals from November 2011 to December 2015 were enrolled in the study. The details of this study have been published previously [12]. This study was approved by the institutional review board of Chonnam National University Hospital (CNUH-2011-172) on 27 October 2011. The Institutional Review Board at each participating hospital approved the study protocol. Patients provided written informed consent to participate in this study. Among 13,104 patients, 1945 with missing values for symptom onset time, hospital arrival time, Thrombolysis In Myocardial Infarction (TIMI) flow grade, and 2-year follow-up loss were excluded (Figure 1). A total of 11,159 patients were categorized into four groups according to the presenting symptoms and history of diabetes (atypical DM, atypical non-DM, typical DM, and typical non-DM).

### 2.2. Definition and Data Collection

Diagnosis of AMI was based on the presenting symptoms, serial ECG changes suggesting myocardial infarction (MI), and an increase in cardiac biomarkers, especially cardiac troponins above the 99th percentile of the upper reference limit. ST-segment elevation myocardial infarction (STEMI) was defined as new ST-segment elevation in more than two contiguous leads, measuring > 0.2 mV in leads V 1–3 or 0.1 mV in other leads, or a new left bundle branch block on 12-lead ECG with at least one positive cardiac troponin T or I. Non-STEMI (NSTEMI) was defined as at least one positive cardiac biomarker without ST-segment elevation. Chronic kidney disease (CKD) was defined as eGFR < 60 mL/min/1.73 m^2^. The features of chest pain were classified as typical or atypical chest pain according to the guidelines for the management of patients with non-ST-elevation acute coronary syndromes [13].

Data were recorded using a Web-based case report form in the Clinical Data Management System (iCReaT) of the Korea NIH by the physicians and trained coordinators. Demographic and baseline characteristics, including cardiovascular risk factors, other comorbidities, initial presenting symptoms, time of presentation, hospital arrival time, balloon time, and laboratory findings were collected. Angiographic findings and details of procedural characteristics, echocardiographic results, and medications were recorded. Treatment strategy and prescription of medications were based on the physician’s discretion. Patients were treated with the standard of care according to the guidelines. Patients were followed up through out-patient clinic visits or by telephonic interviews. Death, MI, and repeat revascularization were assessed at discharge and during the clinical follow-up period.

### 2.3. Study Outcomes

The primary endpoint was defined as patient-oriented composite endpoint (POCE) at 2 years including all-cause death, any MI, and any revascularization. The secondary endpoint was a device-oriented composite endpoint (DOCE) at 2 years including cardiac death, target vessel MI, and target lesion revascularization (TLR). Other secondary endpoints included individual components of POCE and DOCE. Death was considered as cardiac death if the apparent non-cardiac cause of death did not exist. Target vessel MI was defined as the presence of ischemic symptoms or ECG changes with an elevation of cardiac biomarker above the upper limit of normal related to a previously treated target vessel. TLR was defined as repeated revascularization of the target lesion.

### 2.4. Statistical Analysis

Continuous variables are expressed as mean ± SD or median (interquartile range) and assessed using the Wilcoxon signed-rank test or two-sample *t*-test. Categorical variables are expressed as frequencies (percentage) and compared with the chi-square test or Fisher’s exact test. Two-year Kaplan-Meier survival curves of four groups were compared using the log-rank test. The hazard ratios of study outcomes were evaluated using Cox proportional hazards models in the unadjusted and adjusted models. Multivariable-adjust proportional hazard model included age, sex, body mass index, hypertension, chronic kidney disease, dyslipidemia, current smoker, clinical diagnosis, systolic blood pressure, heart rate, Killip class ≥ 2, symptom to door time, and door to balloon time. Potential effect modification by sex, age (≥75 years), diabetes, CKD, clinical diagnosis, multivessel disease, and left ventricular ejection fraction (≥40%) was evaluated through the stratified analysis and interaction testing using a likelihood ratio test. Statistical analyses were performed using SAS 9.4 version (SAS Institute, Cary, NC, USA). Statistical significance was considered when *p* < 0.05.

## 3. Results

### 3.1. Baseline Characteristics

Among 11,161 patients, 1386 (12.4%) patients had atypical chest pain and 3103 (27.8%) had DM. More patients with atypical chest pain were females and older in both DM and non-DM groups (Table 1). Those with atypical chest pain had more cardiovascular risk factors, such as hypertension, CKD, previous history of MI, and old cerebrovascular accident. Dyspnea was more frequently associated with atypical chest pain. A higher proportion of NSTEMI, Killip class ≥ 2, lower left ventricular ejection fraction, longer symptom to door time, and longer door to balloon time was observed in patients with atypical chest pain. Beta-blockers, renin-angiotensin system inhibitors, and statins were less frequently prescribed in patients with atypical chest pain, regardless of diabetic status.

### 3.2. Angiographic and Procedural Characteristics

Patients with atypical chest pain showed more prevalence of multivessel disease in both DM and non-DM groups (Table 2). No differences were seen in the lesion type and treatment strategy between the groups. Post TIMI 3 flow was achieved more often among patients with typical chest pain.

### 3.3. Clinical Outcomes

Median follow-up duration was 730 days (interquartile range 687–759 days). Overall in-hospital death occurred in 388 (3.5%) patients (Table 3). In-hospital death was significantly higher in patients with atypical chest pain among both DM and non-DM groups (11.6% vs. 3.8%, *p* < 0.0001; 7.8% vs. 2.3%, *p* < 0.0001, respectively). The incidences of POCE and all-cause death at 2-year were 17.2% and 9.7%, respectively. The 2-year rate of POCE and DOCE were about twice higher among patients with atypical pain than among those with typical chest pain. All-cause death and cardiac death rate were the main contributing factors of the difference. The 2-year cumulative incidence of all-cause death was highest in the atypical DM group (29.5%) among the four groups. In the logistic regression analysis, patients in the atypical DM group were at the highest risk of POCE (hazard ratio (HR) 1.76, 95% confidence interval (CI) 1.48–2.10), all-cause death (HR 2.23, 95% CI 1.80–2.76), and any MI (HR 2.34, 95% CI 1.51–3.64) in adjusted models (Table 4), as well as DOCE (HR 2.16, 95% CI 1.72–2.72), cardiac death (HR 2.27, 95% CI 1.75–2.94), and target vessel MI (HR 2.17, 95% CI 1.05–4.49). All estimated HRs for clinical endpoints were statistically significant (*p* for trend < 0.0001) among the four groups. The cumulative incidences of primary and secondary clinical endpoints were significantly different (log-rank *p* < 0.0001) (Figure 2 and Figure 3). Atypical chest pain was consistently worse in the subgroup analysis of interest for POCE (Figure 4). Patients with multivessel disease were associated with worse clinical outcomes (*p*-value for interaction = 0.0042).

## 4. Discussion

The main purpose of this study was to evaluate the long-term clinical impact of atypical chest pain among patients with AMI. The key findings were as follows:Patients with atypical chest pain were older, affected more females, had more comorbidities, delayed admission, and delayed treatment.Patients with atypical chest pain showed higher in-hospital death and long-term mortality irrespective of diabetes history.The coexistence of atypical chest pain and DM was at the highest risk of cardiovascular events.

To differentiate typical chest pain from atypical or noncardiac origin is challenging, because the value of chest pain characteristics is limited. Although typical features of chest pain are well described [3,13], a previous study demonstrated that typical symptoms, such as left anterior chest pain and resting pain or radiating pain to the left arm had no value in the diagnosis of AMI. In contrast, atypical symptoms, including radiating pain to the right arm or both arms, vomiting, and observed sweating were more likely to help diagnose AMI [4]. In the extensive review of studies between 1970 and 2005, radiation to right arm or shoulder showed higher likelihood ratio of 4.7 (95% CI 1.9–12) for the diagnosis of AMI than did radiation to the left arm (HR 2.3, 95% CI 1.7–3.1) [14]. Therefore, the likelihood of AMI should not only be based upon history taking but also on a combination of cardiac biomarkers, electrocardiogram, and imaging study [14,15].

Several studies reported the in-hospital mortality for patients with atypical chest pain in the ACS setting. In the National Registry of Myocardial Infarction 2 (NRMI-2) including 1674 hospitals in the United States from June 1994 to March 1998, 142,445 (33%) of 434,877 patients diagnosed with MI complained of no chest pain on presentation. The rate of in-hospital mortality was 23.3% among patients without chest pain compared with 9.3% among those with chest pain [5]. In NRMI-2 registry, 25.3% patients without chest pain and 74.0% patients with chest pain received thrombolysis or primary angioplasty. The Global Registry of Acute Coronary Events (GRACE) study involving 14 countries enrolled 20,881 ACS patients [6]. Patients with atypical symptoms experienced higher in-hospital death (13%) than those with typical symptoms (4.3%). The Gulf Registry of Acute Coronary Events (Gulf RACE) with 6704 ACS patients from six Middle Eastern countries also demonstrated that the absence of typical chest pain was a significant predictor of in-hospital mortality (odds ratio 2.0, 95% CI 1.29–2.75) [16]. Considering the temporal trends of in-hospital mortality that gradually decreased from 4.8% to 3.8% between 2005 and 2018 in KAMIR registry [17], the in-hospital mortality rate (9.2%) in patients with atypical chest pain is still very high, despite patients undergoing treatment with newer drugs and devices than those in the previous decade of NRMI-2 and GRACE study.

There is a lack of evidence on whether atypical chest pain affects long-term clinical outcomes. SWEDEHEART registry included 172,981 AMI patients between 1996 and 2010 [7]. Patients without chest pain were defined as those not having chest pain, including dyspnea and non-specific symptoms. The 5-year mortality in patients without chest pain aged < 65 years was 25.5% compared with 7.9% in those with chest pain. Among patients aged ≥ 65 years, the mortality rates were 57.7% in patients without chest pain and 34% in those with chest pain during 5-year follow-up. In this registry, a total of 40.1% of patients (67,700/168,981) received either percutaneous coronary intervention or coronary artery bypass graft. Of all the patients without chest pain, only 19.3% (4240/21,991) received cardiac intervention. In MONICA/KORA MI registry (population-based registry) from Augsburg, Germany consisting of 1646 patients (1231 men and 415 women aged 25 to 74 years) during median 4.1 years follow-up (interquartile range 15 years), the absence of chest symptoms significantly increased the risk of mortality (HR 1.85, 95% CI 1.13–3.03) [18]. One of the main limitations of this study was that a large number of patients (45.9%) were excluded because of impossible interview (*n* = 660) and incomplete data (*n* = 366) among 2672 patients. Although our study has a relatively shorter follow-up period than did the SWEDEHEART registry and MONICA/KORA MI registry, the strength of this study is that all the patients were treated with percutaneous coronary intervention, and most clinical events were clearly identified. In addition, hard endpoints, such as death and MI were monitored and captured by the government insurance system.

In this study, the combination of atypical chest pain and diabetes history showed the highest incidence of cardiovascular events. These worst outcomes can be explained by the adverse effect of diabetes itself and diabetic autonomic neuropathy. Approximately one-third of type 2 diabetic patients are known to experience serious cardiovascular complications, such as coronary heart disease, MI, and stroke [19,20]. Further, the mortality risk of a patient with a history of DM was comparable to that of a patient with a history of MI or stroke [21]. Meanwhile, diabetic autonomic neuropathy is one of the common complications, resulting in inappropriate perception of pain [22]. Meta-analysis has found that diabetic cardiovascular autonomic neuropathy with ≥2 abnormalities was associated with a 3.45-fold increase in the risk of mortality (95% CI 2.66–4.47) [23]. Accordingly, optimal management of DM and related complications is a major goal to reduce future cardiovascular events.

Modifiable cardiovascular risk factors are substantial targets to reduce mortality. Data from the National Health and Nutrition Examination Survey and Behavioral Risk Factor Surveillance System in the U.S. demonstrated that one third to half of diabetic patients did not reach the target goal of blood sugar level, optimal blood pressure, or low-density lipoprotein (LDL)-cholesterol level [24]. Intensive glycemic control in patients with cardiovascular autonomic neuropathy at baseline had no effect on reducing mortality compared with standard treatment in the ACCORD trial [25]. In contrast, well-treated diabetic patients who maintained target ranges of five risk factors including glycated hemoglobin, LCL-cholesterol, albuminuria, smoking, and blood pressure, had little or no excess risk regarding death, MI, or stroke during median follow-up of 5.7 years [26]. From this point of view, atypical chest pain in diabetic patients could be a potential target to improve clinical outcomes.

Fourth universal definition of MI emphasizes that the use of high-sensitivity cardiac troponin assays is beneficial to rule in/out myocardial injury and to define MI with specific subtypes [15]. Although the diagnostic role of cardiac troponin was established, the detection of a rise and/or fall in cardiac troponin level still needs several hours, resulting in delayed diagnosis of MI in patients with atypical chest pain. Early echocardiographic evaluation combined with troponin assay would be helpful to discriminate vague symptoms. Further detailed assessment strategy related to atypical chest pain is needed. Besides, promising cardiac biomarkers are awaited for early diagnosis of AMI [27].

### Limitations

This study has several limitations. First, this is a prospective, observational multicenter registry, not a randomized controlled design. However, it is difficult to randomize therapeutic strategy according to the presenting symptom because there may be an ethical problem to delay interventional treatment. Second, only 20 tertiary hospitals participated in this registry, and this would not represent the overall population in our country. Third, the duration of diabetes and detailed medications were not analyzed in this study. Fourth, propensity score-matched analysis was not considered because most of the baseline characteristics were heterogeneous between the typical and atypical chest pain groups.

## 5. Conclusions

Atypical chest pain was significantly associated with mortality among patients with AMI. Among four groups, atypical DM group showed the worst 2-year clinical outcomes regarding POCE, all-cause death, and any MI at 2 years. Application of rapid rule in/out AMI protocols would be beneficial to improve clinical outcomes in patients with atypical chest pain.

## Figures and Tables

**Figure 1 jcm-09-00505-f001:**
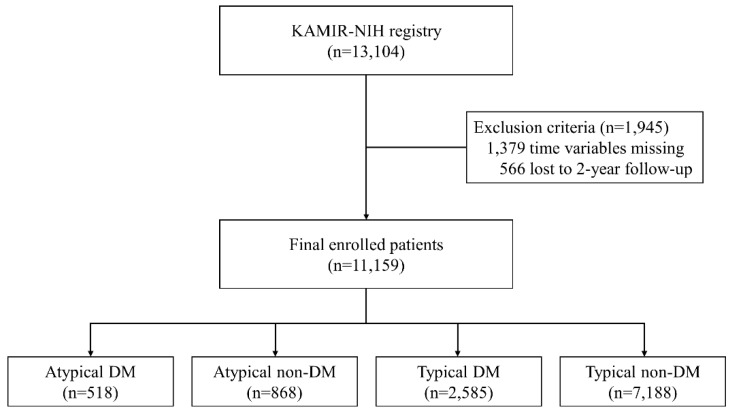
Study flowchart. KAMIR-NIH, Korea Acute Myocardial Infarction-National Institute of Health; DM, diabetes mellitus.

**Figure 2 jcm-09-00505-f002:**
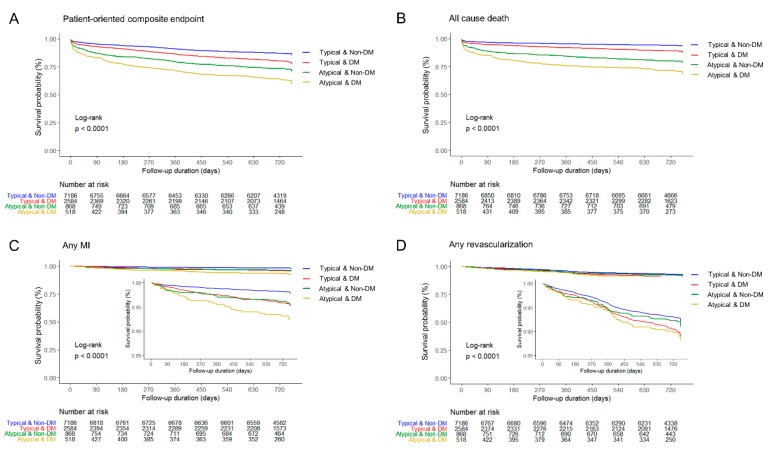
Two-year patient-oriented clinical endpoint and individual clinical outcomes. (**A**) Patient-oriented composite endpoint, (**B**) All-cause death, (**C**) Any myocardial infarction, and (**D**) Any revascularization. DM, diabetes mellitus; MI, myocardial infarction.

**Figure 3 jcm-09-00505-f003:**
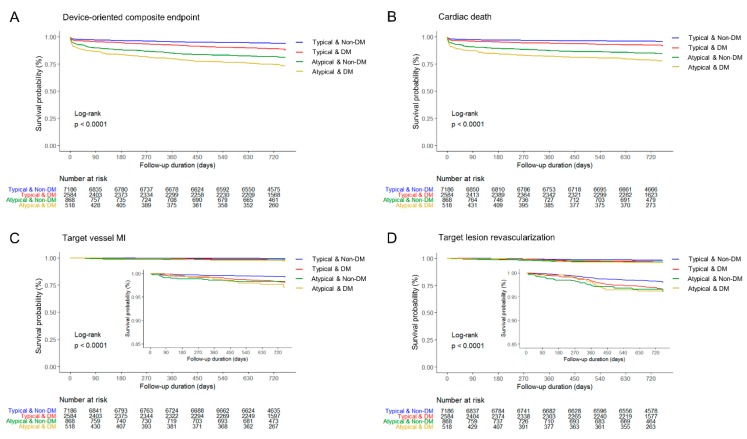
Two-year device-oriented clinical endpoint and individual clinical outcomes. (**A**) Device-oriented composite endpoint, (**B**) Cardiac death, (**C**) Target vessel myocardial infarction, and (**D**) Target lesion revascularization.DM, diabetes mellitus; MI, myocardial infarction.

**Figure 4 jcm-09-00505-f004:**
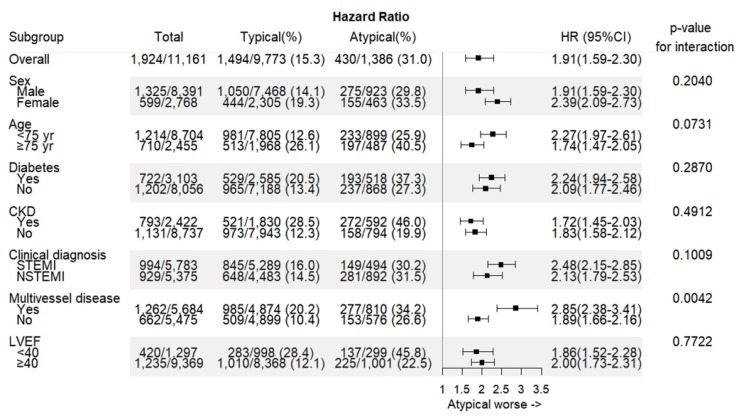
The primary outcome in the subgroup of interest. CKD, chronic kidney disease; STEMI, ST-segment elevation myocardial infarction; LVEF, left ventricular ejection fraction.

**Table 1 jcm-09-00505-t001:** Baseline characteristics.

	Total *n* = 11,161	Diabetic Patients			Non-Diabetic Patients		
		Atypical *n* = 518	Typical *n* = 2585	*p*-Value	Atypical *n* = 868	Typical *n* = 7188	*p*-Value
Age (year)	63.6 ± 12.5	68.5 ± 10.7	65.2 ± 11.2	<0.0001	68.4 ± 12.8	62.0 ± 12.7	<0.0001
Male	8812 (75.2)	328 (63.3)	1822 (70.5)	0.0013	595 (68.6)	5646 (78.6)	<0.0001
Height (cm)	164.4 ± 11.3	161.9 ± 11.4	163.5 ± 12.1	0.0062	162.8 ± 9.0	165.1 ± 11.2	<0.0001
Weight (kg)	65.7 ± 12.1	62.2 ± 12.0	65.8 ± 11.6	<0.0001	61.6 ± 12.2	66.4 ± 12.1	<0.0001
BMI (kg/m^2^)	24.1 ± 3.3	23.5 ± 3.6	24.3 ± 3.3	<0.0001	23.1 ± 3.5	24.2 ± 3.3	<0.0001
**Medical history**							
Hypertension	5894 (50.3)	371 (71.6)	1719 (66.5)	0.0232	429 (49.4)	3095 (43.1)	0.0004
CKD	2566 (21.9)	294 (56.8)	812 (31.4)	<0.0001	298 (34.3)	1018 (14.2)	<0.0001
Dyslipidemia	1330 (11.3)	49 (9.5)	417 (16.1)	0.0001	52 (6.0)	753 (10.5)	<0.0001
Previous MI	824 (7.0)	72 (13.9)	252 (9.8)	0.0048	52 (6.0)	401 (5.6)	0.6187
Old CVA	777 (6.6)	72 (13.9)	234 (9.1)	0.0007	83 (9.6)	355 (4.9)	<0.0001
Current smoker	4713 (40.2)	135 (26.1)	852 (33.0)	0.0021	279 (32.1)	3211 (44.7)	<0.0001
Associated dyspnea	2630 (22.4)	266 (51.4)	545 (21.1)	<0.0001	373 (43.0)	1298 (18.1)	<0.0001
**Clinical Diagnosis**							
STEMI	6078 (51.8)	165 (31.9)	1273 (49.3)	<0.0001	329 (37.9)	4016 (55.8)	<0.0001
NSTEMI	5646 (48.2)	353 (68.2)	1312 (50.8)		539 (62.1)	3171 (44.1)	
Systolic BP (mmHg)	129.9 ± 30.1	124.9 ± 33.4	130.4 ± 29.6	0.0006	124.1±33.9	130.8 ± 29.4	<0.0001
Diastolic BP (mmHg)	78.7 ± 18.5	74.5 ± 20.5	77.7 ± 17.4	0.0008	75.4 ± 21.2	79.7 ± 18.3	<0.0001
HR, bpm	78.1 ± 19.3	85.7 ± 24.6	80.0 ± 19.2	<0.0001	80.8 ± 25.1	76.5 ± 17.9	<0.0001
Hemoglobin (g/dL)	13.9 ± 2.1	12.0 ± 2.4	13.4 ± 2.1	<0001	13.1 ± 2.3	14.3 ± 1.9	<0001
Glucose (mg/dL)	169.7 ± 81.0	245.0 ± 123.8	223.1 ± 99.1	0.0002	157.7 ± 70.4	146.3 ± 53.9	<0001
Creatinine (mg/dL)	1.1 ± 1.1	2.0 ± 2.1	1.3 ± 1.4	<0001	1.3 ± 1.4	1.0 ± 0.8	<0001
Peak CK-MB (ng/mL)	118.80 ± 170.90	62.11 ± 108.80	97.32 ± 134.30	<0001	100.30 ± 164.30	132.80 ± 184.60	<0001
Peak Troponin I (ng/mL)	50.58 ± 109.70	32.31 ± 59.37	49.86 ± 89.21	<0001	45.77 ± 127.60	52.90 ± 116.80	0.1337
Total cholesterol (mg/dL)	179.4 ± 45.7	158.3 ± 46.3	167.2 ± 46.2	0.0001	175.1 ± 45.9	185.8 ± 44.0	<0001
Triglyceride (mg/dL)	136.1 ± 119.4	118.2 ± 70.5	142.1 ± 115.0	<0001	109.7 ± 68.1	138.3 ± 127.4	<0001
HDL-cholesterol (mg/dL)	42.6 ± 11.7	39.0 ± 12.7	41.1 ± 11.6	0.0010	42.6 ± 12.6	43.4 ± 11.4	0.1126
LDL-cholesterol (mg/dL)	113.1 ± 39.2	94.6 ± 39.0	101.9 ± 38.3	0.0004	111.0 ± 40.6	118.5 ± 38.2	<0001
**Killip class**				<0.0001			<0.0001
1	9280 (79.2)	248 (48.0)	1993 (77.1)		521 (60.0)	6054 (84.2)	
2	959 (8.2)	67 (13.0)	241 (9.3)		105 (12.1)	509 (7.1)	
3	789 (6.7)	124 (24.0)	192 (7.4)		142 (16.4)	291 (4.1)	
4	696 (5.9)	78 (15.1)	159 (6.2)		100 (11.5)	334 (4.7)	
Killip class ≥2	2445 (20.9)	270 (52.1)	592 (22.9)	<0.0001	347 (40.0)	1134 (15.8)	<0.0001
LVEF, %	52.0 ± 10.9	46.3±13.0	51.1 ± 11.4	<0.0001	49.6 ± 11.5	53.0 ± 10.2	<0.0001
STD time (minute)	216.0 (87.0, 749.5)	408.5 (92.0, 2880.0)	238.0 (100.5, 822.0)	<0.0001	361.0 (112.0, 1448.0)	194.0 (80.0, 597.0)	<0.0001
DTB time (minute)	104.0 (58.0, 861.0)	401.5 (96.0, 1388.0)	122.0 (59.00, 922.0)	<0.0001	233.5 (75.0, 1125.0)	85.0 (55.0, 728.0)	<0.0001
**Discharge medication**							
Aspirin	11,698 (99.8)	511 (98.7)	2579 (99.8)	0.0024	866 (99.8)	7176 (99.8)	0.6572
P2Y12 inhibitor	11,108 (99.5)	511 (98.7)	2575 (99.6)	0.0146	863 (99.4)	7159 (99.6)	0.4068
CCB	651 (5.8)	53 (10.2)	174 (6.7)	0.0052	53 (6.1)	371 (5.2)	0.2391
Beta blocker	9355 (83.8)	388 (74.9)	2190 (84.7)	<0.0001	655 (75.5)	6122 (85.2)	<0.0001
RAS inhibitor	8882 (79.6)	380 (73.4)	2050 (79.3)	0.0027	640 (73.7)	5812 (80.9)	<0.0001
Statin	10,277 (92.1)	405 (78.2)	2342 (90.6)	<0.0001	748 (86.2)	6782 (94.4)	<0.0001

Values are *n* (%), mean ± SD or median (interquartile range); BMI, body mass index; CKD, chronic kidney disease; MI, myocardial infarction; CVA, cerebrovascular accident; STEMI, ST segment elevation myocardial infarction; NSTEMI, non-ST segment elevation myocardial infarction; BP, blood pressure; HR, heart rate; CK-MB, creatinine kinase-myocardial band; HDL, high-density lipoprotein; LDL, low-density lipoprotein; LVEF, left ventricular ejection fraction; STD, symptom to door; DTB, door to balloon; CCB, calcium channel blocker; RAS, renin angiotensin system.

**Table 2 jcm-09-00505-t002:** Angiographic and procedural characteristics.

	Total *n* = 11,161	Diabetic Patients			Non-Diabetic Patients		
		Atypical *n* = 518	Typical *n* = 2585	*p*-Value	Atypical *n* = 868	Typical *n* = 7188	*p*-Value
**Disease extent**				0.0020			<0.0001
1-vessel disease	5475 (49.1)	175 (33.8)	1054 (40.8)		401 (46.2)	3845 (53.5)	
2-vessel disease	3812 (34.2)	201 (38.8)	978 (37.8)		305 (35.1)	2328 (32.4)	
3-vessel disease	1872 (16.8)	142 (27.4)	553 (21.4)		162 (18.7)	1015 (14.1)	
Multivessel disease	5684 (50.9)	343 (66.2)	1531 (59.2)	0.0030	467 (53.8)	3343 (46.5)	<0.0001
**Culprit lesion**				0.0367			0.0003
Left main	264 (2.4)	24 (4.6)	70 (2.7)		27 (3.1)	143 (2.0)	
LAD	5208 (46.7)	235 (45.4)	1174 (45.4)		357 (41.1)	3442 (47.9)	
LCX	1950 (17.5)	73 (14.1)	452 (17.5)		157 (18.1)	1268 (17.6)	
RCA	3737 (33.5)	186 (35.9)	889 (34.4)		327 (37.7)	2335 (32.5)	
**Lesion type**				0.0521			0.1673
A	148 (1.3)	12 (2.3)	25 (1.0)		10 (1.2)	101 (1.4)	
B1	1337 (12.0)	66 (12.7)	308 (11.9)		103 (11.2)	860 (12.0)	
B2	4180 (37.5)	197 (38.0)	961 (37.2)		299 (34.5)	2723 (37.9)	
C	5494 (49.2)	243 (46.9)	1291 (49.9)		456 (52.5)	3504 (48.8)	
**Lesion treatment**				0.9000			0.4315
Stent	10,343 (93.1)	473 (91.8)	2372 (92.0)		800 (92.9)	6698 (93.6)	
Balloon angioplasty	766 (6.9)	42 (8.2)	206 (8.0)		61 (7.1)	457 (6.4)	
Total number of stents	1.5 ± 0.8	1.6 ± 0.9	1.6 ± 0.8	0.0745	1.5 ± 0.8	1.4 ± 0.7	0.0030
GP IIb/IIIa inhibitor	1720 (15.4)	51 (9.9)	355 (13.7)	0.0166	109 (12.6)	1205 (16.8)	0.0015
Thrombolysis	2790 (25.0)	62 (12.0)	582 (22.5)	<0.0001	143 (16.5)	2003 (27.9)	<0.0001
**Pre TIMI**				0.0062			0.0117
0	5258 (47.1)	172 (33.2)	1066 (41.2)		387 (44.6)	3633 (50.5)	
1	1232 (11.0)	77 (14.9)	322 (12.5)		100 (11.5)	733 (10.2)	
2	1739 (15.6)	109 (21.0)	457 (17.7)		140 (16.1)	1033 (14.4)	
3	2930 (26.3)	160 (30.9)	740 (28.6)		241 (27.8)	1789 (24.9)	
Pre TIMI 0, 1	6490 (58.2)	249 (48.1)	1388 (53.7)	0.0193	487 (56.1)	4366 (60.7)	0.0084
Pre TIMI 3	2930 (26.3)	160 (30.9)	740 (28.6)	0.3006	241 (27.8)	1789 (24.9)	0.0652
**Post TIMI**				0.0028			0.0256
0	43 (0.4)	4 (0.8)	8 (0.3)		8 (0.9)	23 (0.3)	
1	48 (0.4)	8 (1.5)	8 (0.3)		5 (0.6)	27 (0.4)	
2	301 (2.7)	17(3.28)	72 (2.79)		28 (3.2)	184 (2.6)	
3	10,767 (96.5)	489 (94.4)	2497 (96.6)		827 (95.3)	6954 (96.7)	
Post TIMI 0, 1	91 (0.8)	12 (2.3)	16 (0.6)	0.0009	13 (1.5)	50 (0.7)	0.0113
Post TIMI 3	10,767 (96.5)	489 (94.4)	2497 (96.6)	0.0167	827 (95.3)	6954 (96.7)	0.0244

Values are *n* (%). LAD, left anterior descending; LCX, left circumflex; RCA, right coronary artery; GP, glycoprotein; TIMI, Thrombolysis in Myocardial Infarction.

**Table 3 jcm-09-00505-t003:** In-hospital death and 2-year clinical outcomes.

	Total *n* = 11,161	Diabetic Patients			Non-Diabetic Patients		
		Atypical *n* = 518	Typical *n* = 2585	*p*-Value	Atypical *n* = 868	Typical *n* = 7188	*p*-Value
In-hospital death	388 (3.5)	60 (11.6)	98 (3.8)	<0.0001	67 (7.8)	163 (2.3)	<0.0001
Cardiac death	329 (3.0)	50 (9.7)	79 (3.1)	<0.0001	56 (6.5)	144 (2.0)	<0.0001
Non-cardiac death	59 (0.5)	10 (1.9)	19 (0.7)	0.0200	11 (1.3)	19 (0.3)	0.0002
Two-year clinical outcomes							
POCE	1924 (17.2)	193 (37.3)	529 (20.5)	<0.0001	237 (27.3)	965 (13.4)	<0.0001
All-cause of death	1077 (9.7)	153 (29.5)	294 (11.4)	<0.0001	177 (20.4)	453 (6.3)	<0.0001
Any MI	300 (2.7)	29 (5.6)	104 (4.0)	0.1062	30 (3.5)	137 (1.9)	0.0025
Any revascularization	836 (7.5)	43 (8.3)	238 (9.2)	0.5120	60 (6.9)	495 (6.9)	0.9772
DOCE	1020 (9.1)	129 (24.9)	292 (11.3)	<0.0001	159 (18.3)	440 (6.1)	<0.0001
Cardiac death	755 (6.8)	109 (21.0)	206 (8.0)	<0.0001	130 (15.0)	310 (4.3)	<0.0001
Target vessel MI	117 (1.1)	10 (1.9)	43 (1.7)	0.6685	13 (1.5)	51 (0.7)	0.0135
TLR	246 (2.2)	15 (2.9)	78 (3.0)	0.8822	27 (3.1)	126 (1.8)	0.0056

POCE, patient-oriented composite endpoint; MI, myocardial infarction; DOCO, device-oriented composite endpoint; TLR, target lesion revascularization.

**Table 4 jcm-09-00505-t004:** Hazard ratio estimates from logistic regression analysis of the associations for 2-year clinical outcomes.

	Unadjusted Models		Adjusted Models	
	HR(95% CI)	*p* for Trend	HR (95% CI)	*p* for Trend
**POCE**		<0.0001		<0.0001
Typical & Non-DM	1		1	
Typical & DM	1.58 (1.43–1.76)		1.27 (1.13–1.42)	
Atypical & Non-DM	2.24 (1.94–2.58)		1.43 (1.22–1.68)	
Atypical & DM	3.30 (2.83–3.85)		1.76 (1.47–2.10)	
**All-cause of death**		<0.0001		<0.0001
Typical & Non-DM	1		1	
Typical & DM	1.85 (1.60–2.14)		1.44 (1.23–1.69)	
Atypical & Non-DM	3.50 (2.94–4.16)		1.62 (1.33–1.97)	
Atypical & DM	5.36(4.46-6.44)		2.23 (1.80–2.77)	
**Any MI**		<0.0001		<0.0001
Typical & Non-DM	1		1	
Typical & DM	2.20 (1.70–2.84)		1.79 (1.37–2.34)	
Atypical & Non-DM	2.03 (1.37–3.02)		1.67 (1.11–2.53)	
Atypical & DM	3.60 (2.41–5.37)		2.34 (1.51–3.63)	
**Any revascularization**		<0.0001		<0.0001
Typical & Non-DM	1		1	
Typical & DM	1.40 (1.20–1.63)		1.25 (1.06–1.48)	
Atypical & Non-DM	1.13 (0.86–1.47)		1.11 (0.84–1.46)	
Atypical & DM	1.50 (1.10–2.05)		1.34 (0.96–1.86)	
**DOCE**		<0.0001		<0.0001
Typical & Non-DM	1		1	
Typical & DM	1.90 (1.64–2.20)		1.48 (1.26–1.74)	
Atypical & Non-DM	3.26 (2.72–3.91)		1.76 (1.44–2.17)	
Atypical & DM	4.70 (3.87–5.72)		2.16 (1.72–2.72)	
**Cardiac death**		<0.0001		<0.0001
Typical & Non-DM	1		1	
Typical & DM	1.88 (1.58–2.25)		1.49 (1.22–1.81)	
Atypical & Non-DM	3.71 (3.02–4.55)		1.70 (1.34–2.15)	
Atypical & DM	5.46 (4.39–6.80)		2.27 (1.75–2.94)	
**Target vessel MI**		<0.0001		<0.0001
Typical & Non-DM	1		1	
Typical & DM	2.43 (1.62–3.65)		1.97 (1.28–3.01)	
Atypical & Non-DM	2.37 (1.29–4.35)		1.92 (1.02–3.63)	
Atypical & DM	3.30 (1.68–6.51)		2.17 (1.05–4.48)	
**TLR**		<0.0001		<0.0001
Typical & Non-DM	1		1	
Typical & DM	1.79 (1.35–2.38)		1.51 (1.12–2.03)	
Atypical & Non-DM	2.01 (1.32–3.04)		1.79 (1.15–2.77)	
Atypical & DM	2.05 (1.20–3.50)		1.55 (0.88–2.73)	

HR, hazard ratio; CI, confidence interval; POCE, patient-oriented composite endpoint; MI, myocardial infarction; DOCO, device-oriented composite endpoint; TLR, target lesion revascularization; DM, diabetes mellitus. Model adjusted for age, sex, body mass index, hypertension, chronic kidney disease, dyslipidemia, current smoker, clinical diagnosis, systolic blood pressure, heart rate, Killip class ≥2, symptom to door time, door to balloon time.

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
