# Peer review of "Clinical Impact of Atypical Chest Pain and Diabetes Mellitus in Patients with Acute Myocardial Infarction from Prospective KAMIR-NIH Registry"

_jcm, 2020, doi:10.3390/jcm9020505_

Round 1
Reviewer 1 Report
The submitted manuscript is very interesting and presents an important scientific content. The topic and obtained results are important for everyday clinical practice and they can be useful in indicating patients at especially high risk of unfavorable clinical outcomes.
The Introduction briefly presents background of the study.
Methods are described clearly, precisely and understandably.
Results are presented clearly, mostly in tables. However, at this point, I have some questions:
In the Table 1 - what is the meaning of dyslipidemia? The number of such patients is relatively low, while the vast majority of study participants were treated with statin. Is it not attaining the lipid goals despite active treatment? This should be clarified. CKD was highly prevalent, especially among patients with DM. Do you have any information regarding albuminuria level in study participants? It could be important information (or limitation). I agree that it is widely assumed that DM can be considered as important MI risk factor, similarly to the history of MI. However, well treated DM (control of glucose, blood lipids, blood pressure, no smoking and no albuminuria) is associated with even lower risk of MI compared to patients without DM, which was documented by Rawshani A et al. N Engl J Med 2018; 379: 633-644. It could be also mentioned in the discussion.Despite these small remarks, I consider your manuscript very valuable from clinical point of view, indicating patients at especially high risk of CVD event and/or death, and is surely worth to be published in JCM.
Author Response
Dear reviewer
Thank you for your valuable comments and recommendations. Your advice made this revised manuscript to be clear for several issues you addressed.
In the Table 1 –
1. What is the meaning of dyslipidemia? The number of such patients is relatively low, while the vast majority of study participants were treated with statin. Is it not attaining the lipid goals despite active treatment? This should be clarified.
Answer:
Thank you for your valuable comments and advice. All participating centers received study protocol and electronic manual, which clearly defines all the variables in this prospective registry.
According to this manual, ‘dyslipidemia’ was defined ‘Yes’ when the patient had a history of dyslipidemia or prescribed medication for dyslipidemia. To clarify this concern, I corrected the word ‘Risk factors’ to ‘Medical history’ in Table 1.
Moreover, I rechecked this issue from one of the references ‘Kim, J.H. et al. Circ J 2016;80:1427-1436’. The incidence rate of dyslipidemia seems to be different from each other (Table 1). In this table 1, the incidence rate seems to slightly increase from 9.5% (KAMIR, Nov 2005-Oct 2010) to 11.2% (KAMIR-NIH, Nov 2011-Oct 2015).
2. CKD was highly prevalent, especially among patients with DM. Do you have any information regarding albuminuria level in study participants? It could be important information (or limitation).
Answer:
Thank you for your comment and I agree your opinion. Unfortunately, this registry has no information regarding albuminuria level. Therefore, I am sorry that I cannot provide any data for this issue.
3. I agree that it is widely assumed that DM can be considered as important MI risk factor, similarly to the history of MI. However, well treated DM (control of glucose, blood lipids, blood pressure, no smoking and no albuminuria) is associated with even lower risk of MI compared to patients without DM, which was documented by Rawshani A et al. N Engl J Med 2018; 379: 633-644. It could be also mentioned in the discussion.
Answer:
Thank you for your advice. Your recommended article is really helpful to strengthen this manuscript. I added some sentences with this reference in the discussion section (page 11, line 272-275).
In contrast, well-treated diabetic patients who maintained target ranges of five risk factors including glycated hemoglobin, LCL-cholesterol, albuminuria, smoking, and blood pressure) had little or no excess risk regarding death, MI, or stroke during median follow-up of 5.7 years.

Reviewer 2 Report
I feel the manuscript is well written, however, references are VERY old. page 3 line 104.....there is a newer gudeline in 2014 to reference. page 11 line 270, there must be something more recent?
The introduction was based on old references.
I would update references and any newer information. You need to strengthen what's in the literature otherwise, nice paper.
Author Response
Dear reviewer
Thank you for your valuable comments and recommendations. Your advice made this revised manuscript to be clear for several issues you addressed.
1. page 3 line 104.....there is a newer gudeline in 2014 to reference.
Answer:
Thank you for checking references. I fully agree with your opinion. This reference was replaced with the 2014 ACC/AHA guideline. This registry started in Nov 2011. Therefore, I thought that it would be inappropriate to cite the 2014 ACC/AHA guideline which was not yet published at that time.
The features of chest pain were classified as typical or atypical chest pain according to the guidelines for the management of patients with non-ST-elevation acute coronary syndromes [13].
2. page 11 line 270, there must be something more recent?
Answer:
I appreciate your detailed advice. To address this issue, I added some sentences which emphasize the importance of modifiable risk factors with updated new reference (N Engl J Med 2018;379:633-644).
In contrast, well-treated diabetic patients who maintained target ranges of five risk factors including glycated hemoglobin, LCL-cholesterol, albuminuria, smoking, and blood pressure) had little or no excess risk regarding death, MI, or stroke during median follow-up of 5.7 years.
3. The introduction was based on old references. I would update references and any newer information. You need to strengthen what's in the literature otherwise, nice paper.
Answer:
Thank you for your comment. I agree and understand your recommendation. That’s why I was interested in this ‘atypical chest pain’ in AMI patients. When I searched and reviewed published articles from Pubmed, I could find only 2 articles regarding long-term clinical outcomes of atypical chest pain in AMI patients (Reference 7, SWEDEHEART registry published in 2018; Reference 18, MONICA/KORA MI registry from Germany published in 2012). Several articles cited in this manuscript only focused on in-hospital mortality. Therefore, I definitely wanted to know and report this issue from our registry data. Accordingly, I feel difficulty to update with recent literature in the introduction section. I hope you understand why the introduction section is based on relative old references.
